# Learning to Discover
# Efficient Mathematical Identities

**Wojciech Zaremba**
Dept. of Computer Science
Courant Institute
New York Unviersity

**Karol Kurach**
Google Zurich &
Dept. of Computer Science
University of Warsaw

**Rob Fergus**
Dept. of Computer Science
Courant Institute
New York Unviersity

## Abstract

In this paper we explore how machine learning techniques can be applied to the discovery of efficient mathematical identities. We introduce an attribute grammar framework for representing symbolic expressions. Given a grammar of math operators, we build trees that combine them in different ways, looking for compositions that are analytically equivalent to a target expression but of lower computational complexity. However, as the space of trees grows exponentially with the complexity of the target expression, brute force search is impractical for all but the simplest of expressions. Consequently, we introduce two novel learning approaches that are able to learn from simpler expressions to guide the tree search. The first of these is a simple $n$-gram model, the other being a recursive neural-network. We show how these approaches enable us to derive complex identities, beyond reach of brute-force search, or human derivation.

## 1   Introduction

Machine learning approaches have proven highly effective for statistical pattern recognition problems, such as those encountered in speech or vision. However, their use in symbolic settings has been limited. In this paper, we explore how learning can be applied to the discovery of mathematical identities. Specifically, we propose methods for finding computationally efficient versions of a given target expression. That is, finding a new expression which computes an identical result to the target, but has a lower complexity (in time and/or space).

We introduce a framework based on attribute grammars [14] that allows symbolic expressions to be expressed as a sequence of grammar rules. Brute-force enumeration of all valid rule combinations allows us to discover efficient versions of the target, including those too intricate to be discovered by human manipulation. But for complex target expressions this strategy quickly becomes intractable, due to the exponential number of combinations that must be explored. In practice, a random search within the grammar tree is used to avoid memory problems, but the chance of finding a matching solution becomes vanishingly small for complex targets.

To overcome this limitation, we use machine learning to produce a search strategy for the grammar trees that selectively explores branches likely (under the model) to yield a solution. The training data for the model comes from solutions discovered for simpler target expressions. We investigate several different learning approaches. The first group are $n$-gram models, which learn pairs, triples etc. of expressions that were part of previously discovered solutions, thus hopefully might be part of the solution for the current target. We also train a recursive neural network (RNN) that operates within the grammar trees. This model is first pretrained to learn a continuous representation for symbolic expressions. Then, using this representation we learn to predict the next grammar rule to add to the current expression to yield an efficient version of the target.

Through the use of learning, we are able to dramatically widen the complexity and scope of expressions that can be handled in our framework. We show examples of (i) O $\left(n^3\right)$ target expressions which can be computed in O $\left(n^2\right)$ time (e.g. see Examples 1 & 2), and (ii) cases where naive eval-

uation of the target would require *exponential* time, but can be computed in $O(n^2)$ or $O(n^3)$ time. The majority of these examples are too complex to be found manually or by exhaustive search and, as far as we are aware, are previously undiscovered. All code and evaluation data can be found at https://github.com/kkurach/math_learning.

In summary our contributions are:

- A novel grammar framework for finding efficient versions of symbolic expressions.
- Showing how machine learning techniques can be integrated into this framework, and demonstrating how training models on simpler expressions can help which the discovery of more complex ones.
- A novel application of a recursive neural-network to learn a continuous representation of mathematical structures, making the symbolic domain accessible to many other learning approaches.
- The discovery of many new mathematical identities which offer a significant reduction in computational complexity for certain expressions.

---

**Example 1:** Assume we are given matrices $A \in \mathbb{R}^{n \times m}$, $B \in \mathbb{R}^{m \times p}$. We wish to compute the target expression: `sum(sum(A*B))`, i.e. : $\sum_{n,p} AB = \sum_{i=1}^{n} \sum_{j=1}^{m} \sum_{k=1}^{p} A_{i,j} B_{j,k}$ which naively takes $O(nmp)$ time. Our framework is able to discover an efficient version of the formula, that computes the same result in $O(n(m+p))$ time: `sum((sum(A, 1) * B)', 1)`. Our framework builds *grammar trees* that explore valid compositions of expressions from the grammar, using a *search strategy*. In this example, the naive strategy of randomly choosing permissible rules suffices and we can find another tree which matches the target expression in reasonable time. Below, we show trees for (i) the original expression and (ii) the efficient formula which avoids the use of a matrix-matrix multiply operation, hence is efficient to compute.

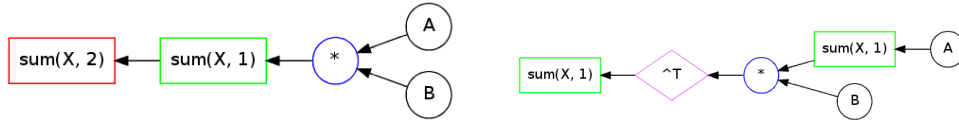

---
**Example 2:** Consider the target expression: `sum(sum((A*B)`$^k$`))`, where $k = 6$. For an expression of this degree, there are 9785 possible grammar trees and the naive strategy used in Example 1 breaks down. We therefore *learn* a search strategy, training a model on successful trees from simpler expressions, such as those for $k = 2, 3, 4, 5$. Our learning approaches capture the common structure within the solutions, evident below, so can find an efficient $O(nm)$ expression for this target:

$k = 2$: `sum((((((sum(A, 1)) * B) * A) * B)'), 1)`

$k = 3$: `sum(((((((((sum(A, 1)) * B) * A) * B) * A) * B)'), 1)`

$k = 4$: `sum((((((((((((sum(A, 1)) * B) * A) * B) * A) * B) * A) * B)'), 1)`

$k = 5$: `sum(((((((((((((((sum(A, 1)) * B) * A) * B) * A) * B) * A) * B) * A) * B)'), 1)`

$k = 6$: `sum(((((((((((((((((sum(A, 1) * B) * A) * B) *A) * B) * A) * B)* A) * B) * A) * B)'), 1)`

## 1.1 Related work

The problem addressed in this paper overlaps with the areas of theorem proving [5, 9, 11], program induction [18, 28] and probabilistic programming [12, 20]. These domains involve the challenging issues of undecidability, the halting problem, and a massive space of potential computation. However, we limit our domain to computation of polynomials with fixed degree $k$, where undecidability and the halting problem are not present, and the space of computation is manageable (i.e. it grows exponentially, but not super-exponentially). Symbolic computation engines, such as Maple [6] and Mathematica [27] are capable of simplifying expressions by collecting terms but do not explicitly seek versions of lower complexity. Furthermore, these systems are rule based and do not use learning approaches, the major focus of this paper. In general, there has been very little exploration of statistical machine learning techniques in these fields, one of the few attempts being the recent work of Bridge *et al.* [4] who use learning to select between different heuristics for 1st order reasoning. In contrast, our approach does not use hand-designed heuristics, instead learning them automatically from the results of simpler expressions.

| Rule | Input | Output | Computation | Complexity |
|------|-------|--------|-------------|------------|
| Matrix-matrix multiply | $X \in \mathbb{R}^{n \times m}, Y \in \mathbb{R}^{m \times p}$ | $Z \in \mathbb{R}^{n \times p}$ | `Z = X * Y` | $O(nmp)$ |
| Matrix-element multiply | $X \in \mathbb{R}^{n \times m}, Y \in \mathbb{R}^{n \times m}$ | $Z \in \mathbb{R}^{n \times m}$ | `Z = X .* Y` | $O(nm)$ |
| Matrix-vector multiply | $X \in \mathbb{R}^{n \times m}, Y \in \mathbb{R}^{m \times 1}$ | $Z \in \mathbb{R}^{n \times n}$ | `Z = X * Y` | $O(nm)$ |
| Matrix transpose | $X \in \mathbb{R}^{n \times m}$ | $Z \in \mathbb{R}^{m \times n}$ | `Z = X`$^T$ | $O(nm)$ |
| Column sum | $X \in \mathbb{R}^{n \times m}$ | $Z \in \mathbb{R}^{n \times 1}$ | `Z = sum(X,1)` | $O(nm)$ |
| Row sum | $X \in \mathbb{R}^{n \times m}$ | $Z \in \mathbb{R}^{1 \times m}$ | `Z = sum(X,2)` | $O(nm)$ |
| Column repeat | $X \in \mathbb{R}^{n \times 1}$ | $Z \in \mathbb{R}^{n \times m}$ | `Z = repmat(X,1,m)` | $O(nm)$ |
| Row repeat | $X \in \mathbb{R}^{1 \times m}$ | $Z \in \mathbb{R}^{n \times m}$ | `Z = repmat(X,n,1)` | $O(nm)$ |
| Element repeat | $X \in \mathbb{R}^{1 \times 1}$ | $Z \in \mathbb{R}^{n \times m}$ | `Z = repmat(X,n,m)` | $O(nm)$ |

Table 1: The grammar $\mathcal{G}$ used in our experiments.

The attribute grammar, originally developed in 1968 by Knuth [14] in context of compiler construction, has been successfully used as a tool for design and formal specification. In our work, we apply attribute grammars to a search and optimization problem. This has previously been explored in a range of domains: from well-known algorithmic problems like knapsack packing [19], through bioinformatics [26] to music [10]. However, we are not aware of any previous work related to discovering mathematical formulas using grammars, and learning in such framework. The closest work to ours can be found in [7] which involves searching over the space of algorithms and the grammar attributes also represent computational complexity.

Classical techniques in natural language processing make extensive use of grammars, for example to parse sentences and translate between languages. In this paper, we borrow techniques from NLP and apply them to symbolic computation. In particular, we make use of an $n$-gram model over mathematical operations, inspired by $n$-gram language models. Recursive neural networks have also been recently used in NLP, for example by Luong *et al.* [15] and Socher *et al.* [22, 23], as well as generic knowledge representation Bottou [2]. In particular, Socher *et al.* [23], apply them to parse trees for sentiment analysis. By contrast, we apply them to trees of symbolic expressions. Our work also has similarities to Bowman [3] who shows that a recursive network can learn simple logical predicates.

Our demonstration of continuous embeddings for symbolic expressions has parallels with the embeddings used in NLP for words and sentence structure, for example, Collobert & Weston [8], Mnih & Hinton [17], Turian *et al.* [25] and Mikolov *et al.* [16].

## 2 Problem Statement

**Problem Definition:** We are given a symbolic *target expression* $\mathbb{T}$ that combines a set of variables $\mathcal{V}$ to produce an output $\mathbb{O}$, i.e. $\mathbb{O} = \mathbb{T}(\mathcal{V})$. We seek an alternate expression $\mathbb{S}$, such that $\mathbb{S}(\mathcal{V}) = \mathbb{T}(\mathcal{V})$, but has lower computational complexity, i.e. $O(\mathbb{S}) < O(\mathbb{T})$.

In this paper we consider the restricted setting where: (i) $\mathbb{T}$ is a homogeneous polynomial of degree $k^*$, (ii) $\mathcal{V}$ contains a single matrix or vector $A$ and (iii) $\mathbb{O}$ is a scalar. While these assumptions may seem quite restrictive, they still permit a rich family of expressions for our algorithm to explore. For example, by combining multiple polynomial terms, an efficient Taylor series approximation can be found for expressions involving trigonometric or exponential operators. Regarding (ii), our framework can easily handle multiple variables, e.g. Figure 1, which shows expressions using two matrices, $A$ and $B$. However, the rest of the paper considers targets based on a single variable. In Section 8, we discuss these restrictions further.

**Notation:** We adopt Matlab-style syntax for expressions.

## 3 Attribute Grammar

We first define an *attribute grammar* $\mathcal{G}$, which contains a set of mathematical operations, each with an associated complexity (the attribute). Since $\mathbb{T}$ contains exclusively polynomials, we use the grammar rules listed in Table 1.

Using these rules we can develop trees that combine rules to form expressions involving $\mathcal{V}$, which for the purposes of this paper is a single matrix $A$. Since we know $\mathbb{T}$ involves expressions of degree

$k$, each tree must use $A$ exactly $k$ times. Furthermore, since the output is a scalar, each tree must also compute a scalar quantity. These two constraints limit the depth of each tree. For some targets $\mathbb{T}$ whose complexity is only $\mathrm{O}\left(()\, n^3\right)$, we remove the matrix-matrix multiply rule, thus ensuring that if any solution is found its complexity is at most $\mathrm{O}\left(()\, n^2\right)$ (see Section 7.2 for more details). Examples of trees are shown in Fig. 1. The search strategy for determining which rules to combine is addressed in Section 6.

## 4 Representation of Symbolic Expressions

We need an efficient way to check if the expression produced by a given tree, or combination of trees (see Section 5), matches $\mathbb{T}$. The conventional approach would be to perform this check symbolically, but this is too slow for our purposes and is not amenable to integration with learning methods. We therefore explore two alternate approaches.

### 4.1 Numerical Representation

In this representation, each expression is represented by its evaluation of a randomly drawn set of $N$ points, where $N$ is large (typically 1000). More precisely, for each variable in $\mathcal{V}$, $N$ different copies are made, each populated with randomly drawn elements. The target expression evaluates each of these copies, producing a scalar value for each, so yielding a vector $t$ of length $N$ which uniquely characterizes $\mathbb{T}$. Formally, $t_n = \mathbb{T}(\mathcal{V}_n)$. We call this numerical vector $t$ the *descriptor* of the symbolic expression $\mathbb{T}$. The size of the descriptor $N$, must be sufficiently large to ensure that different expressions are not mapped to the same descriptor. Furthermore, when the descriptors are used in the linear system of Eqn. 5 below, $N$ must also be greater than the number of linear equations. Any expression $\mathbb{S}$ formed by the grammar can be used to evaluate each $\mathcal{V}_n$ to produce another $N$-length descriptor vector $s$, which can then be compared to $t$. If the two match, then $\mathbb{S}(\mathcal{V}) = \mathbb{T}(\mathcal{V})$.

In practice, using floating point values can result in numerical issues that prevent $t$ and $s$ matching, even if the two expressions are equivalent. We therefore use an integer-based descriptor in the form of $\mathbb{Z}_p{}^\dagger$, where $p$ is a large prime number. This prevents both rounding issues as well as numerical overflow.

### 4.2 Learned Representation

We now consider how to learn a continuous representation for symbolic expressions, that is learn a projection $\phi$ which maps expressions $\mathbb{S}$ to $l$-dimensional vectors: $\phi(\mathbb{S}) \to \mathbb{R}^l$. We use a recursive neural network (RNN) to do this, in a similar fashion to Socher *et al.* [23] for natural language and Bowman *et al.* [3] for logical expressions. This potentially allows many symbolic tasks to be performed by machine learning techniques, in the same way that the word-vectors (e.g.[8] and [16]) enable many NLP tasks to be posed a learning problems.

We first create a dataset of symbolic expressions, spanning the space of all valid expressions up to degree $k$. We then group them into clusters of equivalent expressions (using the numerical representation to check for equality), and give each cluster a discrete label $1 \ldots C$. For example, $A, (A^T)^T$ might have label 1, and $\sum_i \sum_j A_{i,j}, \sum_j \sum_i A_{i,j}$ might have label 2 and so on. For $k = 6$, the dataset consists of $C = 1687$ classes, examples of which are show in Fig. 1. Each class is split 80/20 into train/test sets.

We then train a recursive neural network (RNN) to classify a grammar tree into one of the $C$ clusters. Instead of representing each grammar rule by its underlying arithmetic, we parameterize it by a weight matrix or tensor (for operations with one or two inputs, respectively) and use this to learn the *concept* of each operation, as part of the network. A vector $a \in \mathbb{R}^l$, where $l = 30^\ddagger$ is used to represent each input variable. Working along the grammar tree, each operation in $\mathbb{S}$ evolves this vector via matrix/tensor multiplications (preserving its length) until the entire expression is parsed, resulting in a single vector $\phi(\mathbb{S})$ of length $l$, which is passed to the classifier to determine the class of the expression, and hence which other expressions it is equivalent to.

Fig. 2 shows this procedure for two different expressions. Consider the first expression $\mathbb{S} = (A. * A)' * \mathrm{sum}(A, 2)$. The first operation here is $.*$, which is implemented in the RNN by taking the

---

$^\dagger$Integers modulo $p$

$^\ddagger$This was selected by cross-validation to control the capacity of the RNN, since it directly controls the number of parameters in the model.

two (identical) vectors $a$ and applies a weight tensor $W_3$ (of size $l \times l \times l$, so that the output is also size $l$), followed by a rectified-linear non-linearity. The output of this stage is this $\max((W_3 * a) * a, 0)$. This vector is presented to the next operation, a matrix transpose, whose output is thus $\max(W_2 * \max((W_3 * a) * a, 0), 0)$. Applying the remaining operations produces a final output: $\phi(\mathbb{S}) = \max((W_4 * \max(W_2 * \max((W_3 * a) * a, 0), 0)) * max(W_1 * a, 0))$. This is presented to a $C$-way softmax classifier to predict the class of the expression. The weights $W$ are trained using a cross-entropy loss and backpropagation.

```
(((sum((sum((A * (A')), 1)), 2) * ((A * (((sum((A'), 1)) * A')))') * A)        ((A') * ((sum(A, 2) * ((sum((A'), 1)) * (A * (((sum((A'), 1)) * A')))))))
(sum(((sum((A * (A')), 2)) * ((sum((A'), 1)) * (A * ((A') * A)))), 1))          (sum(((A') * ((sum(A, 2)) * ((sum((A'), 1)) * (A * ((A') * A)))))), 2))
(((sum(A, 1)) * (((sum(A, 2)) * (sum(A, 1)))')) * (A * ((A') * A)))            ((((sum(A, 2)) * ((sum((A'), 1)) * A))')  * (A * (((sum((A'), 1)) * A')))
((((sum((sum((A * (A')), 1)), 2) * ((sum((A'), 1)) * (A * ((A') * A)))))')')   (((sum((A'), 1)) * (A * ((A') * ((sum(A, 2)) * ((sum((A'), 1)) * A)))))')
((sum(A, 1)) * (((A') * (A * ((A') * ((sum(A, 2)) * (sum(A, 1)))))))'))        ((((sum((A'), 1)) * A)') * ((sum((A'), 1)) * (A * (((sum((A'), 1)) * A')))))
((sum((sum((A * (A')), 1)), 2) * ((sum((A'), 1)) * (A * ((A') * A))))          (((A * ((A') * ((sum(A, 2)) * ((sum((A'), 1)) * A))))')  * (sum(A, 2)))
(((sum((sum((A * (A')), 1)), 2) * ((sum((A'), 1)) * A)) * ((A') * A))          (((A') * ((sum(A, 2)) * ((sum((A'), 1)) * A))) * (sum(((A') * A), 2)))
```

(a) Class A                                              (b) Class B

Figure 1: Samples from two classes of degree $k = 6$ in our dataset of expressions, used to learn a continuous representation of symbolic expressions via an RNN. Each line represents a different expression, but those in the same class are equivalent to one another.

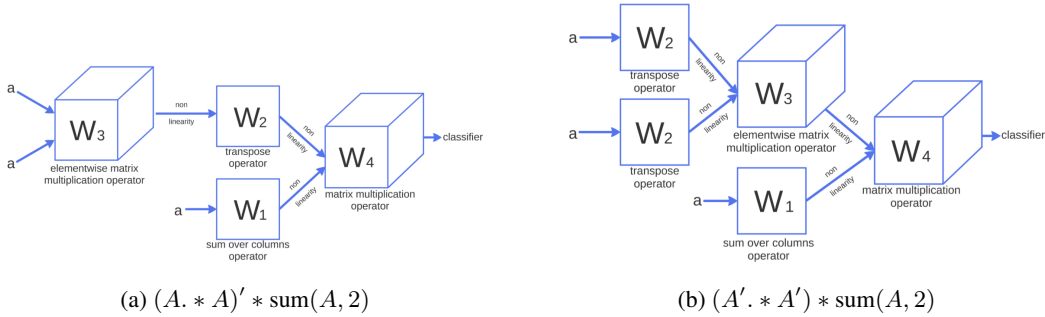

(a) $(A. * A)' * \mathrm{sum}(A, 2)$                     (b) $(A'. * A') * \mathrm{sum}(A, 2)$

Figure 2: Our RNN applied to two expressions. The matrix $A$ is represented by a fixed random vector $a$ (of length $l = 30$). Each operation in the expression applies a different matrix (for single input operations) or tensor (for dual inputs, e.g. matrix-element multiplication) to this vector. After each operation, a rectified-linear non-linearity is applied. The weight matrices/tensors for each operation are shared across different expressions. The final vector is passed to a softmax classifier (not shown) to predict which class they belong to. In this example, both expressions are equivalent, thus should be mapped to the same class.

When training the RNN, there are several important details that are crucial to obtaining high classification accuracy:

- The weights should be initialized to the identity, plus a small amount of Gaussian noise added to all elements. The identity allows information to flow the full length of the network, up to the classifier regardless of its depth [21]. Without this, the RNN overfits badly, producing test accuracies of $\sim 1\%$.
- Rectified linear units work much better in this setting than tanh activation functions.
- We learn using a curriculum [1], starting with the simplest expressions of low degree and slowly increasing $k$.
- The weight matrix in the softmax classifier has much larger ($\times 100$) learning rate than the rest of the layers. This encourages the representation to stay still even when targets are replaced, for example, as we move to harder examples.
- As well as updating the weights of the RNN, we also update the initial value of $a$ (i.e we backpropagate to the input also).

When the RNN-based representation is employed for identity discovery (see Section 6.3), the vector $\phi(\mathbb{S})$ is used directly (i.e. the $C$-way softmax used in training is removed from the network).

## 5    Linear Combinations of Trees

For simple targets, an expression that matches the target may be contained within a single grammar tree. But more complex expressions typically require a linear combination of expressions from different trees.

To handle this, we can use the integer-based descriptors for each tree in a linear system and solve for a match to the target descriptor (if one exists). Given a set of $M$ trees, each with its own integer descriptor vector $f$, we form an $M$ by $N$ linear system of equations and solve it:

$$Fw = t \bmod \mathbb{Z}_p$$

where $F = [f_1, \ldots, f_M]$ holds the tree representations, $w$ is the weighting on each of the trees and $t$ is the target representation. The system is solved using Gaussian elimination, where addition and multiplication is performed modulo $p$. The number of solutions can vary: (a) there can be **no solution**, which means that no linear combination of the current set of trees can match the target expression. If all possible trees have been enumerated, then this implies the target expression is outside the scope of the grammar. (b) There can be **one or more solutions**, meaning that some combination of the current set of trees yields a match to the target expression.

## 6 Search Strategy

So far, we have proposed a grammar which defines the computations that are permitted (like a programming language grammar), but it gives no guidance as to how explore the space of possible expressions. Neither do the representations we introduced help – they simply allow us to determine if an expression matches or not. We now describe how to efficiently explore the space by learning which paths are likely to yield a match.

Our framework uses two components: a **scheduler**, and a **strategy**. The scheduler is fixed, and traverses space of expressions according to recommendations given by the selected strategy (e.g. "Random" or "$n$-gram" or "RNN"). The strategy assesses which of the possible grammar rules is likely to lead to a solution, given the current expression. Starting with the variables $\mathcal{V}$ (in our case a single element $A$, or more generally, the elements $A$, $B$ etc.), at each step the scheduler receives scores for each rule from the strategy and picks the one with the highest score. This continues until the expression reaches degree $k$ and the tree is complete. We then run the linear solver to see if a linear combination of the existing set of trees matches the target. If not, the scheduler starts again with a new tree, initialized with the set of variables $\mathcal{V}$. The $n$-gram and RNN strategies are learned in an incremental fashion, starting with simple target expressions (i.e. those of low degree $k$, such as $\sum_{ij} AA^T$). Once solutions to these are found, they become training examples used to improve the strategy, needed for tackling harder targets (e.g. $\sum_{ij} AA^T A$).

### 6.1 Random Strategy

The random strategy involves no learning, thus assigns equal scores to all valid grammar rules, hence the scheduler randomly picks which expression to try at each step. For simple targets, this strategy may succeed as the scheduler may stumble upon a match to the target within a reasonable time-frame. But for complex target expressions of high degree $k$, the search space is huge and the approach fails.

### 6.2 $n$-gram

In this strategy, we simply count how often subtrees of depth $n$ occur in solutions to previously solved targets. As the number of different subtrees of depth $n$ is large, the counts become very sparse as $n$ grows. Due to this, we use a weighted linear combination of the score from all depths up to $n$. We found an effective weighting to be $10^k$, where $k$ is the depth of the tree.

### 6.3 Recursive Neural Network

Section 4.2 showed how to use an RNN to learn a continuous representation of grammar trees. Recall that the RNN $\phi$ maps expressions to continuous vectors: $\phi(\mathbb{S}) \to \mathbb{R}^l$. To build a search strategy from this, we train a softmax layer on top of the RNN to predict which rule should be applied to the current expression (or expressions, since some rules have two inputs), so that we match the target.

Formally, we have two current branches $b_1$ and $b_2$ (each corresponding to an expression) and wish to predict the root operation $r$ that joins them (e.g. $.*$) from among the valid grammar rules ($|r|$ in total). We first use the previously trained RNN to compute $\phi(b_1)$ and $\phi(b_2)$. These are then presented to a $|r|$-way softmax layer (whose weight matrix $U$ is of size $2l \times |r|$). If only one branch exists, then $b_2$ is set to a fixed random vector. The training data for $U$ comes from trees that give efficient solutions to targets of lower degree $k$ (i.e. simpler targets). Training of the softmax layer is performed by stochastic gradient descent. We use dropout [13] as the network has a tendency to overfit and repeat exactly the same expressions for the next value of $k$. Thus, instead of training on exactly $\phi(b_1)$ and $\phi(b_2)$, we drop activations as we propagate toward the top of the tree (the same

fraction for each depth), which encourages the RNN to capture more local structures. At test time, the probabilities from the softmax become the scores used by the scheduler.

# 7 Experiments

We first show results relating to the learned representation for symbolic expressions (Section 4.2). Then we demonstrate our framework discovering efficient identities. For brevity, the identities discovered are listed in the supplementary material [29].

## 7.1 Expression Classification using Learned Representation

Table 2 shows the accuracy of the RNN model on expressions of varying degree, ranging from $k = 3$ to $k = 6$. The difficulty of the task can be appreciated by looking at the examples in Fig. 1. The low error rate of $\leq 5\%$, despite the use of a simple softmax classifier, demonstrates the effectiveness of our learned representation.

|  | Degree $k = 3$ | Degree $k = 4$ | Degree $k = 5$ | Degree $k = 6$ |
|---|---|---|---|---|
| Test accuracy | $100\% \pm 0\%$ | $96.9\% \pm 1.5\%$ | $94.7\% \pm 1.0\%$ | $95.3\% \pm 0.7\%$ |
| Number of classes | 12 | 125 | 970 | 1687 |
| Number of expressions | 126 | 1520 | 13038 | 24210 |

Table 2: Accuracy of predictions using our learned symbolic representation (averaged over 10 different initializations). As the degree increases tasks becomes more challenging, because number of classes grows, and computation trees become deeper. However our dataset grows larger too (training uses 80% of examples).

## 7.2 Efficient Identity Discovery

In our experiments we consider 5 different families of expressions, chosen to fall within the scope of our grammar rules:

1. $(\sum \mathbf{A}\mathbf{A}^{\mathbf{T}})_{\mathbf{k}}$: $A$ is an $\mathbb{R}^{n \times n}$ matrix. The $k$-th term is $\sum_{i,j}(AA^T)^{\lfloor k/2 \rfloor}$ for even $k$ and $\sum_{i,j}(AA^T)^{\lfloor k/2 \rfloor}A$ , for odd $k$. E.g. for $k = 2 : \sum_{i,j} AA^T$; for $k = 3 : \sum_{i,j} AA^T A$; for $k = 4 : \sum_{i,j} AA^T AA^T$ etc. Naive evaluation is $\mathrm{O}\left(kn^3\right)$.

2. $(\sum(\mathbf{A}.\ast\mathbf{A})\mathbf{A}^{\mathbf{T}})_k$: $A$ is an $\mathbb{R}^{n \times n}$ matrix and let $B = A.\ast A$. The $k$-th term is $\sum_{i,j}(BA^T)^{\lfloor k/2 \rfloor}$ for even $k$ and $\sum_{i,j}(BA^T B)^{\lfloor k/2 \rfloor}$ , for odd $k$. E.g. for $k = 2 : \sum_{i,j}(A.\ast A)A^T$; for $k = 3 : \sum_{i,j}(A.\ast A)A^T(A.\ast A)$; for $k = 4 : \sum_{i,j}(A.\ast A)A^T(A.\ast A)A^T$ etc. Naive evaluation is $\mathrm{O}\left(kn^3\right)$.

3. $\mathbf{Sym}_k$: Elementary symmetric polynomials. $A$ is a vector in $\mathbb{R}^{n \times 1}$. For $k = 1 : \sum_i A_i$, for $k = 2 : \sum_{i<j} A_i A_j$, for $k = 3 : \sum_{i<j<k} A_i A_j A_k$, etc. Naive evaluation is $\mathrm{O}\left(n^k\right)$.

4. $(\mathbf{RBM\text{-}1})_k$: $A$ is a vector in $\mathbb{R}^{n \times 1}$. $v$ is a binary $n$-vector. The $k$-th term is: $\sum_{v \in \{0,1\}^n}(v^T A)^k$. Naive evaluation is $\mathrm{O}\left(2^n\right)$.

5. $(\mathbf{RBM\text{-}2})_k$: Taylor series terms for the partition function of an RBM. $A$ is a matrix in $\mathbb{R}^{n \times n}$. $v$ and $h$ are a binary $n$-vectors. The $k$-th term is $\sum_{v \in \{0,1\}^n, h \in \{0,1\}^n}(v^T Ah)^k$. Naive evaluation is $\mathrm{O}\left(2^{2n}\right)$.

Note that (i) for all families, the expressions yield a scalar output; (ii) the families are ordered in rough order of "difficulty"; (iii) we are not aware of any previous exploration of these expressions, except for $\mathbf{Sym}_k$, which is well studied [24]. For the $(\sum \mathbf{A}\mathbf{A}^{\mathbf{T}})_{\mathbf{k}}$ and $(\sum(\mathbf{A}.\ast\mathbf{A})\mathbf{A}^{\mathbf{T}})_k$ families we remove the matrix-multiply rule from the grammar, thus ensuring that if any solution is found it will be efficient since the remaining rules are at most $\mathrm{O}\left(kn^2\right)$, rather than $\mathrm{O}\left(kn^3\right)$. The other families use the full grammar, given in Table 1. However, the limited set of rules means that if any solution is found, it can at most be $\mathrm{O}\left(n^3\right)$, rather than exponential in $n$, as the naive evaluations would be. For each family, we apply our framework, using the three different search strategies introduced in Section 6. For each run we impose a fixed cut-off time of 10 minutes[§] beyond which we terminate the search. At each value of $k$, we repeat the experiments 10 times with different random initializations and count the number of runs that find an efficient solution. Any non-zero count is deemed a success, since each identity only needs to be discovered once. However, in Fig. 3, we show the fraction of successful runs, which gives a sense of how quickly the identity was found.

---

[§]Running on a 3Ghz 16-core Intel Xeon. Changing the cut-off has little effect on the plots, since the search space grows exponentially fast.

We start with $k = 2$ and increase up to $k = 15$, using the solutions from previous values of $k$ as training data for the current degree. The search space quickly grows with $k$, as shown in Table 3. Fig. 3 shows results for four of the families. We use $n$-grams for $n = 1 \ldots 5$, as well as the RNN with two different dropout rates (0.125 and 0.3). The learning approaches generally do much better than the random strategy for large values of $k$, with the 3-gram, 4-gram and 5-gram models outperforming the RNN.

For the first two families, the 3-gram model reliably finds solutions. These solutions involve repetition of a local patterns (e.g. Example 2), which can easily be captured with $n$-gram models. However, patterns that don't have a simple repetitive structure are much more difficult to generalize. The **(RBM-2)**$_k$ family is the most challenging, involving a double exponential sum, and the solutions have highly complex trees (see supplementary material [29]). In this case, none of our approaches performed better than the random strategy and no solutions were discovered for $k > 5$. However, the $k = 5$ solution was found by the RNN consistently faster than the random strategy ($100 \pm 12$ vs $438 \pm 77$ secs).

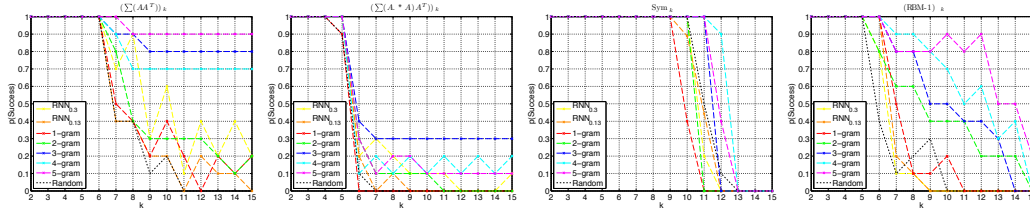

Figure 3: Evaluation on four different families of expressions. As the degree $k$ increases, we see that the random strategy consistently fails but the learning approaches can still find solutions (i.e. p(Success) is non-zero). Best viewed in electronic form.

|  | $k = 2$ | $k = 3$ | $k = 4$ | $k = 5$ | $k = 6$ | $k = 7$ and higher |
|---|---|---|---|---|---|---|
| # Terms $\leq$ O $\left(n^2\right)$ | 39 | 171 | 687 | 2628 | 9785 | Out of memory |
| # Terms $\leq$ O $\left(n^3\right)$ | 41 | 187 | 790 | 3197 | 10k+ | |

Table 3: The number of possible expressions for different degrees $k$.

## 8 Discussion

We have introduced a framework based on a grammar of symbolic operations for discovering mathematical identities. Through the novel application of learning methods, we have shown how the exploration of the search space can be learned from previously successful solutions to simpler expressions. This allows us to discover complex expressions that random or brute-force strategies cannot find (the identities are given in the supplementary material [29]).

Some of the families considered in this paper are close to expressions often encountered in machine learning. For example, dropout involves an exponential sum over binary masks, which is related to the **RBM-1** family. Also, the partition function of an RBM can be approximated by the **RBM-2** family. Hence the identities we have discovered could potentially be used to give a closed-form version of dropout, or compute the RBM partition function efficiently (i.e. in polynomial time). Additionally, the automatic nature of our system naturally lends itself to integration with compilers, or other optimization tools, where it could replace computations with efficient versions thereof.

Our framework could potentially be applied to more general settings, to discover novel formulae in broader areas of mathematics. To realize this, additional grammar rules, e.g. involving recursion or trigonometric functions would be needed. However, this would require a more complex scheduler to determine when to terminate a given grammar tree. Also, it is surprising that a recursive neural network can generate an effective continuous representation for symbolic expressions. This could have broad applicability in allowing machine learning tools to be applied to symbolic computation.

The problem addressed in this paper involves discrete search within a combinatorially large space – a core problem with AI. Our successful use of machine learning to guide the search gives hope that similar techniques might be effective in other AI tasks where combinatorial explosions are encountered.

## Acknowledgements

The authors would like to thank Facebook and Microsoft Research for their support.

## Footnotes

*I.e. It only contains terms of degree $k$. E.g. $ab + a^2 + ac$ is a homogeneous polynomial of degree 2, but $a^2 + b$ is not homogeneous ($b$ is of degree 1, but $a^2$ is of degree 2).

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
