[Reviews · NeurIPS 2014]

Submitted by Assigned_Reviewer_13

First, apologies for the brevity of this review- I had written a version with more detailed comments and had found a few typos, but can't find it now.

This paper is an application of DNNs to a novel area (finding mathematical identities) and my feeling is that while the ideas are novel, this seems like preliminary work that maybe doesn't quite rise to the level of a NIPS publication. Specifically, your method doesn't seem to show particularly impressive performance in discovering identities, even though you only gave it a very limited set of inputs. The expressions that you were finding identities on were from such a limited set that it would probably have been easier to just work out some mathematical rule to discover them rather than applying DNNs; it seems like using a sledgehammer to crack a nut. Of course you might argue that your approach could be extended to a much more general range of expressions, but it's not clear to me that your existing approach would very easily scale out in that way, or that doing so would even make sense given your so-so initial results.

Summary: I do think this is an interesting paper, but I think it's maybe a little but below the threshold for publication.

Submitted by Assigned_Reviewer_19

This paper describes two methods for searching for simpler symbolic expression trees. One of these is based on the use of ngrams computed over sequences of sub-expressions. The other uses a recursive neural network to create continuous space representations of symbolic expression trees. These representations are then used to predict what rule is likely to be used to combine subtrees. The use of ngrams appears from Figure 3 to be better than the RNN on two of the three tasks.

Comments wrt specific review criteria:
Quality - High.
Clarity - The paper is somewhat hard to follow, especially Section 7.2.
Originality - High.
Significance - Low. It is unlikely that anyone seeking to perform efficient computations would ever do this. The problem seems contrived.
Summary: This paper describes ways of searching for more computationally efficient symbolic expression trees. The significance of the methods is unclear.

Submitted by Assigned_Reviewer_41

This paper introduces an interesting solution to finding more efficient algebraic identities. It is very interesting to see that distributed, recursive neural representations can successfully classify these identities.

The problem definition seems quite restricted.

This is not well explained:
"weight tensorW3 (of size lxlxl, so that the output is also size l),"
Does this mean you use a neural tensor network instead of a simple linear network?
Bowman showed that the tensor network works a lot better in these than the RNN.

7.2 Efficient Identity Discovery
It is unclear if you're still using the RNN here for classification but you are evaluation different search strategies.
I didn't quite follow how you generate proposals in your search. This could be explained better.

typos:
has contains a single matrix
listed in table Table 1.
vector is the presented to the next operation
The final vector is pass to a softmax classifier
as to how explore space of possible
The scheduler is is fixed
The difficulty of the task by be appreciated by looking
which can easily captured with
Should be RNN, not TNN in figure 3.
Summary: This paper introduces an interesting solution to finding more efficient algebraic identities. It is very interesting to see that distributed, recursive neural representations can successfully classify these identities.

Submitted by Assigned_Reviewer_43

The paper introduces a novel use case for neural nets, which is learning a representation for symbolic expressions and finding their mathematical identities.
Even though the expressions studied are very limited and there is no proof that they can be extended to more complicated expressions, I find the paper interesting. And I believe it could have good impact for symbolic software libraries.

-The problem statement states that we're looking for expressions with lower computational complexity. But nowhere in the algorithm such a constraint is enforced. I understand that this is not trivial, but have authors taught about possible solutions to limit the search space to such solutions?

-About the restrictions mentioned in section 2, is the scheduler the only limiting factor for more complicated mathematical expression? I would have liked to see more formal or mention of empirical experiments for setting these restrictions.

-Also the structure of the paper is a not easy to follow. It would be better if stages were explained more clearly.

-Is there a constraint for size of the vector/matrix? e.g. for k=1,n=1 the expression for RBM-1 in the supplementary material doesn't seems to be valid.

Some typos:
332: show for
148: has contains
289: is is
338: by be
Summary: The paper introduced a novel approach for representing symbolic expressions and finding their mathematical identities. Even though it is a limited in the expressions it can handle, it is an interesting and novel approach.
Author Feedback
Author rebuttal: We would like to thank the reviewers for their comments. In particular, we are pleased that all three reviewers recognized the originality of our paper. However, there were some serious misunderstandings that we would like to address:

R13: “The expressions …. would probably have been easier to just work out some mathematical rule to discover them rather than applying DNNs; it seems like using a sledgehammer to crack a nut.”

We respectfully disagree with this and feel that R13 has perhaps misunderstood the difficulty of the problems addressed. In the problems we consider there is simply *no other way* to discover the identities, other than with our learning approaches (n-gram/RNN).

First, they are certainly not derivable by humans, i.e. there is no way to “just work out some mathematical rule”. Perhaps R13 would be convinced of this point if they were to spend a few moments trying to derive one of the identities for themselves, e.g. a “simple” one like (\sum(A.*A)*A)_5. In fact, the problem is deceptively complex, and not so easily derived. Our solution presented in the supplementary material makes it clear that the identities are well beyond anything than can be manually derived. Indeed, for the RBM families considered we in fact worked with mathematicians for several months to try to discover identities for k>=3, but they were not able to make progress (thus spurring the development of techniques presented in our paper). Furthermore, if the identities really are so easy to discover manually, why are they not already known? Fig. 1 shows a simple result, that to the best of our knowledge, is new.

Second, brute force search is totally infeasible. k=6 already requires a week of computation and so k=15 would take thousands of years. Yet our learning approaches can discover solutions for k=15 for some families in a matter of minutes.

R13: “Specifically, your method doesn't seem to show particularly impressive performance in discovering identities”; “or that doing so would even make sense given your so-so initial results.”

We respectfully disagree that our results are not impressive. Given the novelty of the task, there are few reference points with which to gauge good performance. As argued above, the problems addressed are very challenging and not solvable by any existing manual or computational technique. However, the perception of poor performance perhaps arises from two causes:

The first concerns the presentation of results in Fig. 3: just because the curves from the learning approaches are not at 100% for large k, it does not mean that our approach is not successful. On the contrary, even if our learning approaches find a solution a few % of the time, this is resounding success since, for large k, the brute force approaches will essentially never find a solution. We can certainly change the plots to show a binary measure of success instead, so as to make the performance of our methods more apparent.

Second, to be scientifically honest we deliberately chose to show failure cases. As all 3 reviewers acknowledge, our algorithms are a first attempt at trying to solve these type of problems using learning, so naturally do not succeed in all cases and it is important to show this. However they are significantly better than the random search baseline and will hopefully inspire others to develop more sophisticated approaches, so opening up a new application area for ML.

R13: “...a very limited set of inputs. The expressions that you were finding identities on were from such a limited set..”
R41: “The problem definition seems quite restricted.”

Mathematics is a big field and so it is natural that the first attempt should adopt some kind of restriction in the type of expressions. This can either be in terms of “depth”, i.e. complexity of expressions or “breadth”, i.e. diversity of expressions. In our paper we chose to focus on depth, which is the harder of the two as the search space is exponentially larger. We constrained the breadth somewhat by considering polynomial expressions but we respectfully disagree that this constitutes a “very limited set”. First, they are sufficiently rich to be impossible to solve with brute force approaches (or manual manipulation). Second, the identities discovered have clear practical utility. The RBM identities can be used to give a novel way of computing RBM partition functions, quite different from existing sampling approaches. The other families are useful for linear algebra computation and can be integrated into Matlab or compilers (see below).

By adding to the dictionary of grammar rules we expect to be able to handle a greater breadth of expressions, such as trigonometric identities, recursive identities (e.g. FFT), multinomial sums and Fibonacci family identities. Furthermore, trivial changes are needed to discover tensor identities, which are particularly difficult to manipulate manually.

R19: “Significance - Low. It is unlikely that anyone seeking to perform efficient computations would ever do this. The problem seems contrived.”

We respectfully disagree. Compilers routinely replace computation with equivalent operations that are more efficient. Indeed, there are entire frameworks built to do this, including ones that focus on matrix operations, such as the Delite framework developed at Stanford (http://stanford-ppl.github.io/Delite/). Of course, the equivalences have to be discovered in the first place, which is where our approach is needed.

R19: “The paper is somewhat hard to follow, especially Section 7.2.”
Thank you. We will rewrite it to improve clarity.

R41: “Does this mean you use a neural tensor network instead of a simple linear network?”
Yes, we use tensor networks for operators taking two operands (e.g. matrix multiplication), and matrices for operators taking a single operand (e.g. transposition).

Also, thanks for the suggestions for clarification & typos - we will fix these.